# Increasing Risk of Tick-Borne Disease through Growth Stages in Ticks

Makoto Kondo *, Yoshiaki Matsushima, Takehisa Nakanishi, Shohei Iida, Habe Koji and Keiichi Yamanaka

Department of Dermatology, Graduate School of Medicine, Mie University, Mie 514-8507, Japan
* Correspondence: kondomak@clin.medic.mie-u.ac.jp

**Abstract:** *Rickettsia* and *Coxiella* spp. are pathogens transmitted by ticks to humans. However, the developmental stage of the tick carrying the greatest risk of infection is unknown. Detection of pathogen-specific genes proves that ticks carrying *Rickettsia* or *Coxiella* spp. constitute a reservoir of infection. However, conventional PCR methods are unable to quantitate the pathogens within ticks. In the present study, we collected ticks in the endemic area of Japanese spotted fever, caused by *Rickettsia japonica*, and determined the rate of tick-borne pathogens carried by the ticks. As a method of evaluation, next-generation sequencing was used to estimate the proportion of pathogens in 10 adult and 10 larval ticks. Ticks were identified *Haemaphysalis longicornis* (H.L) from the results of the sequencing of PCR products amplified using tick identification-specific primers. The gene detection rates were 10/10 for *Rickettsia* sp. and 10/10 for *Coxiella* sp. among the adult ticks. For the larval ticks, the ratios were 7/10 and 5/10 for *Rickettsia* sp. and *Coxiella* sp., respectively. The largest proportion of *Coxiella* sp.-specific DNA reached 96% in one adult tick. The proportion of *Rickettsia* sp. genes ranged from 1.76% to 41.81% (mean, 15.56%) in the adult ticks. The proportions of *Coxiella* and *Rickettsia* spp. genes in the larvae ranged from 0% to 27.4% (mean 5.86%) and from 0% to 14.6% (mean 3.38%), respectively. When the percentage of *Rickettsia* sp., out of all pathogens detected via next-generation sequencing, was analyzed between the adult and larval stages of the ticks, a significant difference was observed at $p = 0.0254$. For *Coxiella* sp., a highly significant difference ($p < 0.0001$) was found between the adult and larval stages of the ticks. In conclusion, the detection rates and proportions of *Rickettsia* and *Coxiella* spp. genes were highest in adult H.L ticks. The risk of contracting tick-borne infections may increase with bites from adult ticks, especially those harboring *Coxiella* sp.

**Keywords:** tick; next-generation sequencing; *Rickettsia*; *Coxiella*

## 1. Introduction

The ticks suck blood by biting. The tick's mouthparts are shaped like scissors, which cut through the skin. The tick then inserts its jagged teeth, which connect with the host and feed from a pool of blood that forms under the skin. During blood sucking, they transmit a variety of pathogens into the host. For example, *Rickettsia* and *Coxiella* are bacterial pathogens carried by ticks that cause serious diseases in humans. Delayed diagnosis results in life-threatening conditions such as disseminated intravascular coagulation and multiple organ failure [1–4]. There are numerous reports on the prevalence of genes specific for these pathogens in ticks [5–9]. However, the growth stage during which bacterial pathogens develop and replicate within the tick has not been determined. Several bacterial pathogens, such as *Borrelia* [10] and *Francisella* [11], along with *Rickettsia* and *Coxiella*, are known to be transmitted by ticks to humans. However, the stage of the tick during which the transmissibility of disease-associated bacteria is most likely remains unknown. This is because, to our knowledge, there have been no reports of ticks being attached to patients who have developed tick-borne diseases. This information will be helpful for disease prevention. In this study, ten adult and ten larval ticks were collected, and next-generation

sequencing (NGS) was performed to determine the growth stages of ticks during which tick-borne diseases are most transmissible.

## 2. Materials and Methods

The types of bacteria in ticks collected from Minami-Ise, Mie Prefecture, Japan, were determined by analyzing the bacterial 16S rRNA regions using NGS. The area where larvae and adult ticks were collected in approximately the same numbers as in the Minami-Ise area, where the distribution of ticks was previously surveyed in summer, was selected as the tick survey area for this survey. This area was known to be an endemic area for Japanese spotted fever (JSF), with a large number of ticks harboring *Rickettsia japonica* (*R. japonica*). Furthermore, in this survey area, the reason for which *Haemophysalis longicornis* (H.L) was collected in large numbers during the summer season was that it was thought to reduce variation in the prevalence of tick-borne pathogens among tick species. Ten adult and ten larval ticks were captured in widely dispersed locations using the flagging method during the summer season. The ticks were placed in bottles and soaked overnight in 70% ethanol. The ticks were then removed using tweezers, crushed separately using a BioMasher II (Nippi, Tokyo, Japan), and stored in new tubes. DNA from the tick was extracted using a QIAamp DNA mini kit (Qiagen, German town, MD, USA) according to the manufacturer's protocol. To identify the bacterial species, NGS analysis of the bacterial 16S rRNA lesions using DNA extracted from the ticks was performed by Macrogen Japan Corp. (Tokyo, Japan). In addition, DNA extracted from the ticks was used to perform polymerase chain reaction (PCR) using primers widely used for tick species identification [12]. The reaction mixture was placed in a thermal cycler (ASTEL GeneAtlas 485, Fukuoka, Japan), where it underwent one cycle of preheating (94 °C, 10 s), 35 cycles of denaturation (94 °C, 10 s), annealing (55 °C, 30 s), extension (72 °C, 30 s), and one cycle of delay (72 °C, 5 min). Sequences of the amplified PCR products were analyzed using Eurofins Genomics (Tokyo, Japan). The number of *Rickettsia* sp. and *Coxiella* sp. genes detected in the DNA extracted from ten adult and ten larval ticks was calculated. To make a comparative estimate of the number of pathogens in the adult and larval ticks, the percentages of *Rickettsia* sp. (KJ619629) and *Coxiella* sp. (KC776318) gene sequences identified were measured when the number of sequences of all pathogens was 100, based on the results of NGS analysis of DNA extracted from the ticks. Two independent groups (adult vs. larva) were compared using Student's *t*-test.

## 3. Results

The ten adult and ten larval ticks were identified as H.L from the results of the sequencing of PCR products amplified using tick identification-specific primers. The numbers of *Rickettsia* and *Coxiella* spp. genes detected in the ten adult and ten larval ticks and the percentages of *Rickettsia* and *Coxiella* among all detected pathogen sequences are shown in Figure 1. *Rickettsia* sp. and *Coxiella* sp. genes were each detected in 10/10 of the adult ticks. Conversely, *Rickettsia* sp. was found in 7/10 and *Coxiella* sp. was found in 5/10 of the larval ticks. *Anaplasma* sp. was detected in only 1/10 of the adult ticks and was not detected in the larval ticks. *Borrelia* sp. and *Francisella* sp. were not detected in either life stage of the ticks. *Coxiella* sp. was the most common species among all the adult ticks (mean, 58.15%). The highest percentage of *Coxiella* sp. genes was approximately 96%, in one adult tick. The percentage of *Rickettsia* sp. genes found in the adult ticks ranged from 1.76% to 41.81% (mean, 15.56%). However, in the larval ticks, *Coxiella* sp. accounted for 0%–27% (mean 5.86%) of the genes detected, while *Rickettsia* sp. accounted for 0%–14.6% (mean 3.38%). Statistically significant differences were observed between adult ticks and larvae for both *Rickettsia* sp. and *Coxiella* sp. (Figure 1).

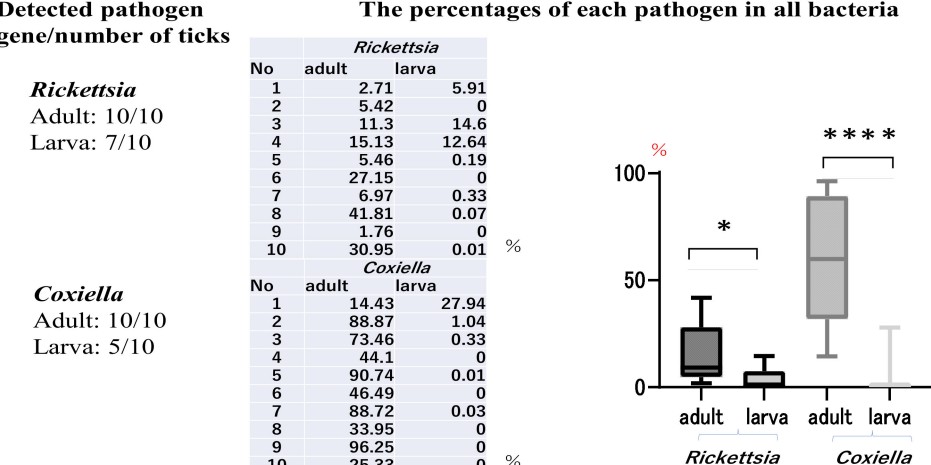

**Figure 1.** The gene detection rates in the adult ticks were 10/10 for *Rickettsia* sp. and 10/10 for *Coxiella* sp. Conversely, the detection rates for larval ticks were 7/10 for *Rickettsia* sp. and 5/10 for *Coxiella* sp. * When the percentage of *Rickettsia* sp., out of all pathogens detected via next-generation sequencing, was analyzed between the adult and larval stages of the ticks, a significant difference was observed at $p = 0.0254$. **** For *Coxiella* sp., a highly significant difference ($p < 0.0001$) was found between the adult and larval stages of the ticks.

*Diplorickettsia* sp., which has not been previously reported in Japan, was detected in an adult tick. Moreover, *Aeromonas* sp. was detected in two larval ticks [13].

## 4. Discussion

Identifying pathogens within a tick does not qualify it as a reservoir of infection or prove infectivity, if the tick cannot transmit the pathogen. Furthermore, the NGS method used in the current study could only analyze the genus level. Within the genera *Rickettsia* and *Coxiella*, the gene sequences are extremely similar; therefore, it is difficult to identify them as individual species, even by quantitative PCR. The area where the ticks were surveyed is one of the most endemic areas in Japan for JSF caused by *R. japonica*. *R. japonica* has been detected in the blood of JSF-infected patients, and ticks in this region can transmit *Rickettsia* to the human body. We recognize the presence of residents with positive anti-IgG antibodies against *Coxiella burnetti* in the area studied (unpublished data). Hence, we considered it worthy of recognition, even at the genus level. Several studies have reported the percentage of *Rickettsia* in ticks at different growth stages [5–8]. The parentage of *Coxiella* sp. specific DNA detected in ticks has also been reported [8,9]. However, the growth stages of ticks during which the risk of transmission is the greatest remain unknown. There are no reports on the development of tick-borne infections caused by ticks still attached to the human body. PCR was used to confirm the presence of pathogens in the ticks. Gene detection indicates that ticks are carrying *Rickettsia* sp. or *Coxiella* sp.; however, evaluating the risk of infection is not feasible because PCR methods are not useful for determining the number of pathogens. This study investigated the detection rates of tick-borne pathogen genes and the relative percentages of these pathogens by performing NGS on ticks at different stages (adults and larvae). NGS results showed that adult ticks contained significantly higher amounts of *Rickettsia* sp. and *Coxiella* sp., whereas larval ticks contained few of these bacteria. The present study was based on a small number of ticks from a limited area. However, the area surveyed had the highest reported cases of JSF in Japan [1,14]. Most reported ticks in this area have been identified as H.L [15]. H.L harbors *R. japonica*, which causes JSF [16]. Therefore, the results regarding *Rickettsia* sp. in this study, which compared 10 adult H.L and 10 larval H.L ticks, should provide credible data on the risk of JSF infection from a bite by an adult H.L tick. *Coxiella* sp. were detected with NGS using DNA extracted from the ticks. This proved that ticks with *Coxiella* sp. that sucked human blood lacked sufficient infection in non-endemic areas of JSF in Mie Prefecture [17].

However, the *Coxiella sp.* in the surveyed area in Mie Prefecture likely had the potential to infect humans because residents in the area had antibody titers against *Coxiella burnetti*. As shown in Figure 2, ticks suck blood from wild animals, grow into adults, and lay eggs. The larvae analyzed in this study contained small amounts of tick-borne pathogens, and it is likely that *Rickettsia* sp. and *Coxiella* sp. multiply in the tick body as it grows. If the wildlife is the source of the pathogens detected in the ticks, then all adult ticks feeding from the same local wildlife likely contain the same types of pathogens and to the same degree. The present study was a comparative assessment of the risk of infection during the adult and larval stages of H.L ticks. In the future, it will be necessary to continue the survey by identifying areas where adult, nymph, and larval ticks can be captured in larger but equal numbers.

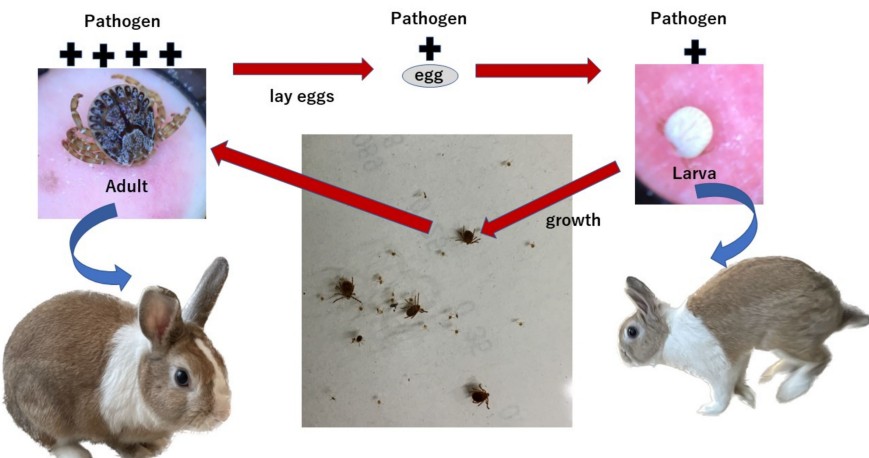

**Figure 2.** As ticks grow, pathogens multiply inside them. Adult ticks suck blood from wild animals and lay eggs, and tick-borne pathogens are present in the eggs. A few tick-borne pathogens are also present in larval ticks that have not yet sucked blood from a wild animal.

## 5. Conclusions

Wild animals serve as reservoirs for tick-borne diseases, and the number of these pathogens in ticks increases as they feed and grow. The number of pathogens transferred from adult ticks in the eggs is low. Larval ticks have not yet sucked blood from wild animals. Thus, the risk of disease pathogen transmission depends on the growth stage of the tick. Adult ticks are more likely to transmit pathogens than larvae.

**Author Contributions:** Conceptualization, M.K. Methodology, Y.M., T.N. and S.I. Writing—original draft preparation, M.K. Writing—review and editing, H.K. and K.Y. All authors have read and agreed to the published version of the manuscript.

**Funding:** This research received no external funding.

**Institutional Review Board Statement:** Not applicable.

**Informed Consent Statement:** Not applicable.

**Data Availability Statement:** All data supporting the findings of this study have been included in this article.

**Conflicts of Interest:** The authors declare no conflict to interest.

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
