# Peer review of "Increasing Risk of Tick-Borne Disease through Growth Stages in Ticks"

_clinpract, doi:10.3390/clinpract13010022_

Round 1
Reviewer 1 Report
Overall, this manuscript is well composed. The introduction is relevant, and theory based. Sufficient information about the previous study findings is presented for readers to follow the present study rationale and procedures. The methods are generally appropriate, although clarification of a few details and provision of a rationale for the use of this method of measuring the number of pathogens should be provided. Overall, the results are clear and compelling with two possible exceptions. The low sample size and the collection methods might have not been used to the author’s advantage. Also, some of the word choices used in this manuscript may have negatively impacted the quality and the merit of this study. The authors make a systematic contribution to the research literature in this area of investigation. Overall, this is a reasonable manuscript that has implications for the tick field, information for practical and clinical setting; however, I strongly believe that this manuscript needs additional works. Specific comments follow.
Title, Abstract, and Introduction
p. 1, line 2: The term "growth stages" suggests that there must be a " must me larvae – nymphs - adults". Be more specific why larvae and adults were examining but not nymphs?
p. 1, line 17: “gene” Be specific – is it the 16rRNA? If a specific region, please provide the sequence or accession number.
p. 1, line 29 – 30: Combine the two sentences or rewrite. Try to fit it into line 27 – 28.
p. 1, line 36: “several pathogens” be more specific and list the references
p. 1, line 27-30: The word "severe disease" repeated too many times. Recommend replacing it with other similar word if possible.
p. 1, line 32-33: What is the purpose of this sentence? This study is not sampling tick attached to human.
Methods
Page 2 line 50 – 51: using 16sRNA to identify tick species or pathogens in the tick. This is not clear. Please consider revise it.
Please provide statical method/s used in this study.
Provide the seasonality and time of year when these samples were collected.
Results
Due the small sample sizes, the author/s should be cautious when analyzing or interpreting these results.
Fig1: (Table?) It is unclear to how the author/s calculate gene detection rate and how does this detection rate relate to number of pathogens. For the graph, “Rickettsia and Coxiella” need to be italicize. Provide statistical analysis method, what kind of test was used the adult and larva comparison? Provide appropriate label for X-axis and Y-axis.
Page 3, Line 77-78: “P = 0.0254 and P<0.0001)” need to be italicize.
If possible, please provide the sequences and chromatogram as part of the result section.
Discussion and Results
At its current form, the discussion section is highly depended on the result of this study. Currently, there are a few major issues need to address before the discussion section can be evaluate.
Author Response
Responses to the comments of Reviewer #1
Comments to the Author:
- 1, line 2: The term "growth stages" suggests that there must be a " must me larvae – nymphs - adults". Be more specific why larvae and adults were examining but not nymphs?
Response: The results of the previous tick survey were used as a guide for this survey. The area was an endemic area for Japanese spotted fever. Many nymphs were collected more than adult and larva ticks. So adult and larva ticks were selected for statistical comparison since the same amount of adult and larva ticks were collected. The reason is mentioned in the Materials and Methods section.
- 1, line 17: “gene” Be specific – is it the 16rRNA? If a specific region, please provide the sequence or accession number.
Response: We supplied accession numbers.
- 1, line 29 – 30: Combine the two sentences or rewrite. Try to fit it into line 27 – 28.
Response: We have compiled and rewritten the two sentences.
- line 36: “several pathogens” be more specific and list the references
Response: We added tick-borne pathogens and added references.
- 1, line 27-30: The word "severe disease" repeated too many times. Recommend replacing it with other similar word if possible.
Response: We avoided the repetition of the same words.
- 1, line 32-33: What is the purpose of this sentence? This study is not sampling tick attached to human.
Response: This is a sentence that makes the purpose of this study difficult to understand and has been omitted.
- Page 2 line 50 – 51: using 16sRNA to identify tick species or pathogens in the tick. This is not clear. Please consider revise it.]
Response: We have clearly indicated that the method used to identify the tick species was qPCR and the method used to identify the pathogen species was NGS.
- Please provide statical method/s used in this study.
Response: We provide statical methods in the Materials and methods section.
- Provide the seasonality and time of year when these samples were collected.
Response: We have described when we captured the ticks in the Materials and methods section.
- Due the small sample sizes, the author/s should be cautious when analyzing or interpreting these results. Fig1: (Table?) It is unclear to how the author/s calculate gene detection rate and how does this detection rate relate to number of pathogens. For the graph, “Rickettsia and Coxiella” need to be italicize. Provide statistical analysis method, what kind of test was used the adult and larva comparison? Provide appropriate label for X-axis and Y-axis.
Response: Thank you for your valuable advice. The detection method is described as a t-test, and the evaluation method of the number of detections is added to the Figure 1 title. The detail of the detection rate was also added in the “Materials and methods” section. We have added a percentage notation on the X-axis and highlighted in brackets that the Y-axis represents Rickettsia and Coxiella.
- Page 3, Line 77-78: “P = 0.0254 and P<0.0001)” need to be italicize.
Response: We changed them in italics.
- If possible, please provide the sequences and chromatogram as part of the result section.
Response: We are sorry. We have not received this information from the company that requested the NGS. We received analysis information only on accession numbers.
- At its current form, the discussion section is highly depended on the result of this study. Currently, there are a few major issues need to address before the discussion section can be evaluate.
Response: We delved into the results of this tick survey and discussed them again. The detection of a tick-borne pathogen in ticks does not necessarily mean that it will cause infection even if injected into the human body. However, the area surveyed is a high-incidence area for Japanese spotted fever, which is transmitted by Rickettsia japonica. It has been proven that the Rickettsia japonica gene can be detected in the human body infected with Japanese spotted fever. Residents of the area also have antibodies against Coxiella. Therefore, it would be significant to evaluate the risk of these pathogens through growth stages in ticks.
Reviewer 2 Report
Response to Authors: Increasing risk of tick-borne disease through growth stages in ticks
While I think the intended aim of this paper is important, the description of methodology is simply not sound for this type of tick study, nor do the results support the conclusions. (e.g. Simply finding pathogens within a tick does not constitute a reservoir of infection or prove infectiousness if the tick is unable to transmit that pathogen. This study did not assess transmission dynamics so cannot claim that the adults in this study were more likely to transmit a pathogen than the larvae).
The authors used NGS to detect bacterial pathogens and calculated how many genes of these genera were detected. How was the number of target genes/rates determined? Also, not all Coxiella or Rickettsia species are pathogenic or transmittable, so I’m confused as to why this study focused on genus level rather than specific species. I think perhaps comparing the tick-borne pathogen quantification/detection ability of qPCR vs NGS would be a more useful investigation.
I recommend continuing the study with a larger sample of ticks and to include nymphs (it is well-documented in the literature that they are the predominant life stage responsible for disease transmission due to having taken a blood meal and their extremely small size). Larvae are rarely infected because it is rare that tickborne illnesses are transmitted transovarially. Larvae would be a good comparison group, but this type of study is invalid without considering nymphs.
Overall, this study requires a deeper dive into the literature, a more concrete objective, and a more honed discussion that fits the results.
Author Response
Responses to the comments of Reviewer #2
Comments to the Author:
- While I think the intended aim of this paper is important, the description of methodology is simply not sound for this type of tick study, nor do the results support the conclusions. (e.g. Simply finding pathogens within a tick does not constitute a reservoir of infection or prove infectiousness if the tick is unable to transmit that pathogen. This study did not assess transmission dynamics so cannot claim that the adults in this study were more likely to transmit a pathogen than the larvae).
Response: Thank you for your valuable comment. The reviewer is right, the detection of a pathogen in a tick does not necessarily guarantee that it will be injected into the human body. It also does not necessarily mean that infection is established. However, the area studied is one of the most endemic areas of Japanese spotted fever caused by Rickettsia japonica in Japan. Therefore, ticks in this area have the ability to inject pathogens into the human body and can also cause infection. The same reason for Coxiella can be said because the residents had IgG antibodies to Coxiella.
- The authors used NGS to detect bacterial pathogens and calculated how many genes of these genera were detected. How was the number of target genes/rates determined? Also, not all Coxiella or Rickettsia species are pathogenic or transmittable, so I’m confused as to why this study focused on genus level rather than specific species. I think perhaps comparing the tick-borne pathogen quantification/detection ability of qPCR vs NGS would be a more useful investigation.
Response: We have mentioned and included the results evaluated by t-test in the “Materials and methods” section. As the reviewer pointed out, it would be a better assessment of the risk of tick pathogens if they could be identified to species. However, the Rickettsia sp. or Coxiella sp. sequences are extremely similar, and it is difficult to identify them as related species by qPCR. The area surveyed is an endemic area for tick-borne infections, and we have determined that the genus level will be sufficient for evaluation. This was added to the discussion.
- I recommend continuing the study with a larger sample of ticks and to include nymphs (it is well-documented in the literature that they are the predominant life stage responsible for disease transmission due to having taken a blood meal and their extremely small size). Larvae are rarely infected because it is rare that tickborne illnesses are transmitted transovarially. Larvae would be a good comparison group, but this type of study is invalid without considering nymphs.
Response: We thought we should have included nymph in our survey. However, we could have previously collected a very large number of nymphs than adult and larva ticks during the same area and summer season, and we decided that the same number of adult and larva ticks should be used for comparison in this survey to reduce statistical bias.
- Overall, this study requires a deeper dive into the literature, a more concrete objective, and a more honed discussion that fits the results.
Response: Reflecting the valuable advice of reviewers, we have specified more specific methods and results for this study. We have developed a discussion based on these results.
Reviewer 3 Report
However, the growth stages of ticks, 82 at which the risk of transmission is greatest, remain unknown.
Can the above statements be better explained? There are already known reports of adult tick transmitting infections.
Author Response
Responses to the comments of Reviewer #3
However, the growth stages of ticks, 82 at which the risk of transmission is greatest, remain unknown.
Can the above statements be better explained? There are already known reports of adult tick transmitting infections.
Response: In this study, NGS results showed that adult ticks harbor more tick-borne pathogens than larva ticks. Ticks suck blood from wildlife and grow. It is likely that rickettsia sp. and Coxiella sp. multiply in the tick's body. If the tick-borne pathogens in ticks are carried in the wildlife, then all adult ticks sucked the wildlife will contain the same kinds of the pathogen to the same degree. This added to the discussion.